

# A full reference quality assessment method with fused monocular and binocular features for stereo images

Xiaojuan Hu, Jinxin Bai, Chunyi Chen and Haiyang Yu

School of Computer Science and Technology, Changchun University of Science and Technology, Changchun, China

## ABSTRACT

Aiming to automatically monitor and improve stereoscopic image and video processing systems, stereoscopic image quality assessment approaches are becoming more and more important as 3D technology gains popularity. We propose a full-reference stereoscopic image quality assessment method that incorporate monocular and binocular features based on binocular competition and binocular integration. To start, we create a three-channel RGB fused view by fusing Gabor filter bank responses and disparity maps. Then, using the monocular view and the RGB fusion view, respectively, we extract monocular and binocular features. To alter the local features in the binocular features, we simultaneously estimate the saliency of the RGB fusion image. Finally, the monocular and binocular quality scores are calculated based on the monocular and binocular features, and the quality scores of the stereo image prediction are obtained by fusion. Performance testing in the LIVE 3D IQA database Phase I and Phase II. The results of the proposed method are compared with newer methods. The experimental results show good consistency and robustness.

# INTRODUCTION

In order to pursue a better visual experience, the images people watch shift from 2D images to 3D images that are more three-dimensional and realistic. Compared with 2D images, 3D images can help people understand the content more accurately. Therefore, stereoscopic images are widely used in many fields such as pattern recognition, movie production, virtual games, high-precision maps, telemedicine, and distance education. The production of stereoscopic images and videos has also become a research hotspot in related fields (*Smolic et al., 2007*). However, various distortions will inevitably occur during the collection, storage, encoding, transmission and compression of stereoscopic images. These distortions directly affect the adequacy and accuracy of information expression, leading to a reduction in the visual quality of stereoscopic images (*Zhai & Min, 2020*). The efficient stereoscopic image quality assessment (SIQA) method is widely used to evaluate the performance of stereoscopic image processing algorithms, such as stereoscopic image compression and denoising. In addition, a good SIQA method can also be used to guide stereo image processing and image realization and optimization processes. Currently, SIQA methods can be divided into two categories according to whether they use human

Corresponding author
Xiaojuan Hu, huxj@cust.edu.cn

subjective perception: subjective methods and objective methods (*Chandler, 2013*). The subjective method can directly reflect the true quality of the image, and the evaluation results are reliable, but it is time-consuming, high-cost, and unsuitable for practical applications. The objective method is based on the human visual system (HVS) and can automatically predict the quality evaluation of stereoscopic images. Therefore, the objective method is more suitable for application in the quality evaluation of actual scenes.

2D image quality assessment (2D-IQA) method has developed relatively maturely in recent years. For example, *Liu et al. (2023)* proposed a multi-scale depth-free image quality assessment method with spatial optimal scale filtering analysis. *Yue et al. (2023)* proposed a method that can measure modified A reference-free quality assessment method for the deviation of facial images from reality. *Varga (2022)* proposed a full reference image quality assessment method based on Grunwald-Letnikov derivative and image gradient. The assessment of the SIQA method is also more complicated due to the unique stereoscopic vision characteristics of stereoscopic images. Similar to two-dimensional image quality assessment, SIQA methods can generally be divided into three categories based on their dependence on the original image, namely full reference SIQA (FR-SIQA) reduced reference SIQA (RR-SIQA) and no reference SIQA (NR-SIQA).

## FR-SIQA

Most early FR-SIQA methods directly applied the 2D-IQA method to independently process each view image to predict the stereoscopic image quality without considering the binocular vision characteristics. However, *Chen et al. (2013)* pointed out that using the 2D-IQA method to predict the quality of stereoscopic images has better results in the case of symmetrical distortion, but poorer results in the case of asymmetrical distortion. Symmetric distortion means that the left and right views have the same degree of distortion, while asymmetric distortion means that the left and right views have different types of distortion or different degrees of distortion. Therefore, people began to consider the characteristics of binocular vision to solve the problem of asymmetric distortion. *Shao et al. (2013)* proposed a new FR-SIQA method which first classifies stereo images into non-corresponding areas, binocular fusion and suppression areas, then extracts quality-sensitive features from reference and distorted stereo images; finally through Each area is evaluated independently taking into account its binocular perceptual characteristics, and all evaluation results are integrated into an overall score. *Fezza & Larabi (2014)* proposed to use local entropy as a weight factor to combine the left and right views to synthesize the middle view, and the performance of this method was significantly improved. *Geng et al. (2016)* used the independent component analysis method to extract the features of stereoscopic images, and combined their feature similarity and local brightness consistency to construct a FR-SIQA. *Ma et al. (2017)* introduced a binocular perception model based on SSIM, which jointly considers the quality of cyclopean images and differential images to predict the perceptual quality of stereoscopic images. *Chen & Zhao (2019)* proposed a full reference method that combines local and global visual features to perceive the quality of stereoscopic images. *Si et al. (2021)* first proposed a SIQA method to divide stereo images into occluded and non-occluded areas. First, the disparity information and the Euclidean

distance between stereo pairs are used to find a segmentation strategy for occluded and non-occluded areas in the scene; the occluded areas represent monocular vision, and the unoccluded areas show binocular vision of HVS. Global and local features are then extracted from the regions and used to predict visual quality. *Ahmed Seghir et al. (2023)* calculated the binocular summation map by adding the left and right images of the stereo pair, and then used an improved distortion pixel measure based on gradient similarity to evaluate the quality of the binocular summation map.

## RR-SIQA

*Yang et al. (2018)* proposed a RR-SIQA method based on sparse coding. *Wan, Gu & Zhao (2019)* proposed a RR-SIQA method that uses sparse representation and natural scene statistics to simulate the brain's visual perception. The distribution statistics of the extracted categorical visual primitives are sparsely represented to measure the visual information, and the natural scene statistics of the local normalized brightness coefficients are used to evaluate the natural loss due to the presence of distortion. The quality score is calculated by calculating the difference between visual information and natural scene statistics for the original and distorted images, and then using a prediction function trained by support vector regression. *Ma, Xu & Han (2021)* proposed a RR-SIQA method based on gradient-based sparse representation and structural degradation. The proposed method is based on two main tasks: the first task extracts distribution statistics of visual primitives through gradient sparse representation, while the second task measures the distribution of visual primitives by extracting joint statistics of gradient magnitude and Laplacian of Gaussian features due to Structural degradation of stereoscopic images caused by distortion.

## NR-SIQA

In current research, NR-SIQA methods can generally be categorized into those based on traditional machine learning and those based on deep learning.

The first category typically relies on natural scene statistics (NSS) features, which are then input into machine learning methods such as support vector regression (SVR) to obtain quality assessment scores. *Messai, Hachouf & Seghir (2020)* introduced a novel NR-SIQA method. They first considered the presence of binocular rivalry, generating an intermediate view from the left and right views, and then extracted features like gradient magnitude, relative gradient direction, and relative gradient magnitude from the intermediate view. They employed AdaBoost regression to achieve the final quality assessment score. *Zhang et al. (2022)* proposed a multi-scale perceptual feature-based SIQA method. Stereoscopic images were transformed using a pyramid to obtain sensitized images of varying resolutions. These images were further processed to extract structural features, intrinsic structure features, and depth structure features. A genetic algorithm-based SVR (GA-SVR) was employed to achieve the final quality score. However, due to limitations such as high complexity and imprecise regression models associated with manually extracted features, these traditional NR-SIQA methods cannot achieve optimal consistency with HVS. *Shen et al. (2023)* proposed a natural scene statistical-based NR-

SIQA method. Firstly, two-dimensional features are extracted from monocular images in spatial and transform domains. Then, perception-related features associated with three-dimensional quality are extracted from stereo visual perception information. Following this, through natural scene statistical modeling and adaptive principal component analysis feature pruning, the optimal regression model is fitted using SVR.

With the advancement of deep learning techniques, research has found that convolutional neural networks (CNN) have significant advantages in addressing computer vision problems. CNN is capable of automatically extracting image features from the deeper layers of the input images, eliminating the need for manual feature extraction, and performing quality regression on the extracted features. For example, *Shi et al. (2020)* proposed a multi-task CNN to simultaneously learn image quality assessment and distortion type recognition, extending a single-column multi-task CNN model to a three-column model, achieving good performance on asymmetrically distorted stereoscopic images. *Chang, Li & Zhao (2020)* proposed a hierarchical multi-scale no-reference SIQA method based on dilated convolution, which employed dilated convolution on newly generated monocular images to simulate multi-scale features of human vision. *Li et al. (2020)* also introduced a three-channel CNN-based SIQA method using three-dimensional visual saliency maps, which were derived from the two-dimensional saliency regions and depth saliency maps of stereo images. *Bourbia, Karine & Chetouani (2021)* proposed a novel network block called the Fusion Weight Allocation Module, which adaptively weighted features to guide the fusion of stereo image features. *Sandić-Stanković, Kukolj & Le Callet (2022)* utilized a general regression neural network to model the relationship between extracted features and subjective scores. While some deep learning-based NR-SIQA methods have achieved promising results in the literature, they often came with high training time complexity, resulting in lengthy processing times for generating results. Thus, accurately assessing the visual perceptual quality of stereoscopic images remained challenging.

Stereoscopic images in the real world often suffer from asymmetrical distortion, where the distortion level and type differ between left and right views. In order to solve the problem of inaccurate SIQA evaluation results caused by asymmetric distortion, researchers have made a series of efforts. Existing binocular perception studies suggest that during binocular integration in the HVS, both the right and left view images are simultaneously perceived, resulting in a fused image in the brain. However, during binocular rivalry in the HVS, only the right or left view is perceived. Therefore, the quality of stereoscopic images is not only related to the distortion level of each individual left or right view but also to the binocular stereoscopic perceptual experience. Inspired by this, this article proposes a monocular-binocular fusion-based SIQA (MB-FR-SIQA) method. The method proposed in this article has the following characteristics:

1. Traditional grayscale fusion views effectively address the issue of asymmetric distortion. We constructed fusion views from the red, green, and blue (RGB) color channels instead of the grayscale channel to ensure observers perceive the same spatial domain.

2. To ensure the integrity of feature extraction, we extracted local phase and local amplitude features based on extracting global features. We used saliency maps to optimize the quality score of local features.

3. When people observe images, both monocular and binocular visual characteristics occur simultaneously. This article extracted monocular and binocular features as quality-sensitive features and computed the final quality scores.

## MATERIALS AND METHODS

In order to solve the problem of inaccurate evaluation caused by asymmetric distortion, we propose an MB-FR-SIQA method to evaluate the quality of stereo images. The framework of the MB-FR-SIQA method is shown in Fig. 1. In the first place, we constructe the RGB fusion view from the given stereo image pair by considering the existence of binocular competition under asymmetric distortion. Then, monocular features are extracted from the left and right views, while binocular features are extracted from the RGB fusion view. When extracting binocular features, not only global features but also local features are extracted to ensure the completeness of feature extraction. Then the saliency map of the RGB fusion view is calculated, and the weights of the local features are adjusted using the saliency map. Finally, the monocular and binocular quality scores are calculated based on the monocular and binocular features, and fused to obtain the final predicted quality scores of the stereo image.

### Build the RGB fusion view

We build RGB fusion views to simulate the brain's fusion of perceptual information from the left and right eyes. However, when there are varying degrees of distortion in stereoscopic image pairs, the visual stimuli received by the left and right eyes at the same retinal position are not equal, resulting in a mismatch in the perceived images and leading to the binocular rivalry phenomenon. Therefore, considering binocular rivalry is essential during the process of constructing fusion views. Given that Gabor filters can model frequency-oriented decomposition in the primary visual cortex and capture energy in a highly localized manner in both spatial and frequency domains (*Field, 1987*). We employ the energy response of Gabor filters to simulate visual stimulus intensity (*Su, Bovik & Cormack, 2011*), thus emulating binocular rivalry.

Constructed the fusion view on a grayscale channel in previous work (*Messai, Hachouf & Seghir, 2020*), and experimental results demonstrated its effectiveness for quality assessment. In this article, the fusion view is constructed on the red, green, and blue (R, G, B) channels to ensure that observers could see the same spatial domain. The RGB fusion view is defined as $C(x, y)_n$, where $n \in \{R, G, B\}$. The calculation formula of the RGB fused view is shown as follows:

$$C(x,y)_n = \omega_l(x,y)_n \times I_l(x,y)_n + \omega_r(x+m,y)_n \times I_r(x+m,y)_n \tag{1}$$

where $(x, y)$ represents the coordinates of the pixel. $n$ is the color channel index. $I_l(x, y)$ and $I_r(x+m, y)$ represent the left and right views, respectively. $m$ is the disparity index

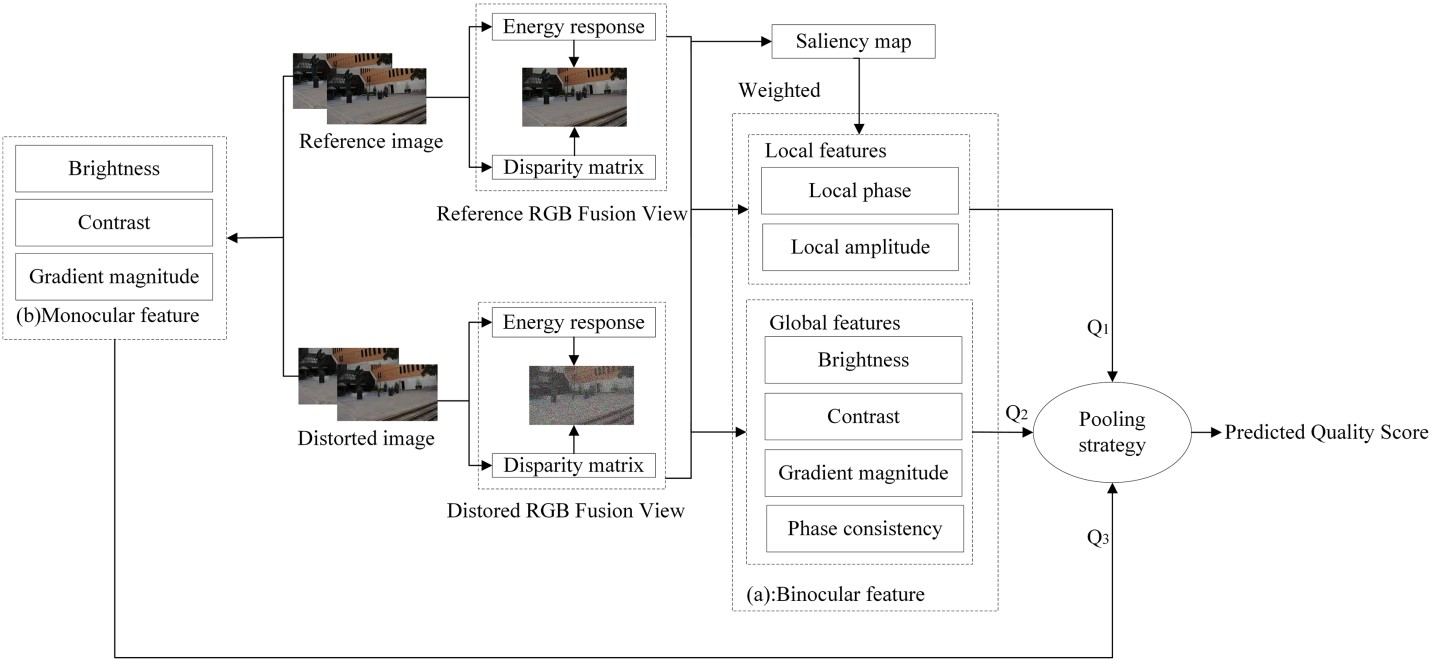

**Figure 1** **The framework of the MB-FR-SIQA method.** (A) Extract binocular features. (B) Extract monocular features. The Reference/Distorted image is from LIVE 3D Image Quality Database—Phase I. The Reference/Distorted RGB fusion view is generated using the author's code.

calculated using the SSIM-based stereo disparity algorithm. $\omega_l(x, y)$ and $\omega_r(x+m, y)$ respectively represent the weight coefficients of the left and right eyes. The weights are given by:

$$\omega_l(x, y) = \frac{G_l(x, y)}{G_l(x, y) + G_r(x + m, y)} \tag{2}$$

$$\omega_r(x + m, y) = \frac{G_r(x + m, y)}{G_l(x, y) + G_r(x + m, y)} \tag{3}$$

where $G_l(x, y)$ and $G_r(x+m, y)$ are the total sums of amplitude responses of the Gabor filter bank in the left and right views, across eight orientations (horizontal, vertical, diagonal). These responses are used to simulate the visual stimulus intensity of the left and right views. We define the Gabor filter bank $G(x, y)$ is calculated as follows:

$$
\begin{aligned}
G(x, y) &= \frac{1}{2\pi\sigma_x\sigma_y} \cdot e^{-\frac{1}{2}\left[(R_1/\sigma_x)^2 + (R_2/\sigma_y)^2\right]} \cdot e^{i(x\zeta_x + y\zeta_y)}, \\
R_1 &= (x - m_x) \cdot \cos(\theta) + (y - m_y) \cdot \sin(\theta), \\
R_2 &= -(x - m_x) \cdot \sin(\theta) + (y - m_y) \cdot \cos(\theta)
\end{aligned}
\tag{4}
$$

where $m_x$ and $m_y$ determine the center of the Gabor filter's receptive field. $\sigma_x$ and $\sigma_y$ are the standard deviations of the elliptical Gaussian envelope along the $x$ and $y$ axes. $\zeta_x$ and $\zeta_y$

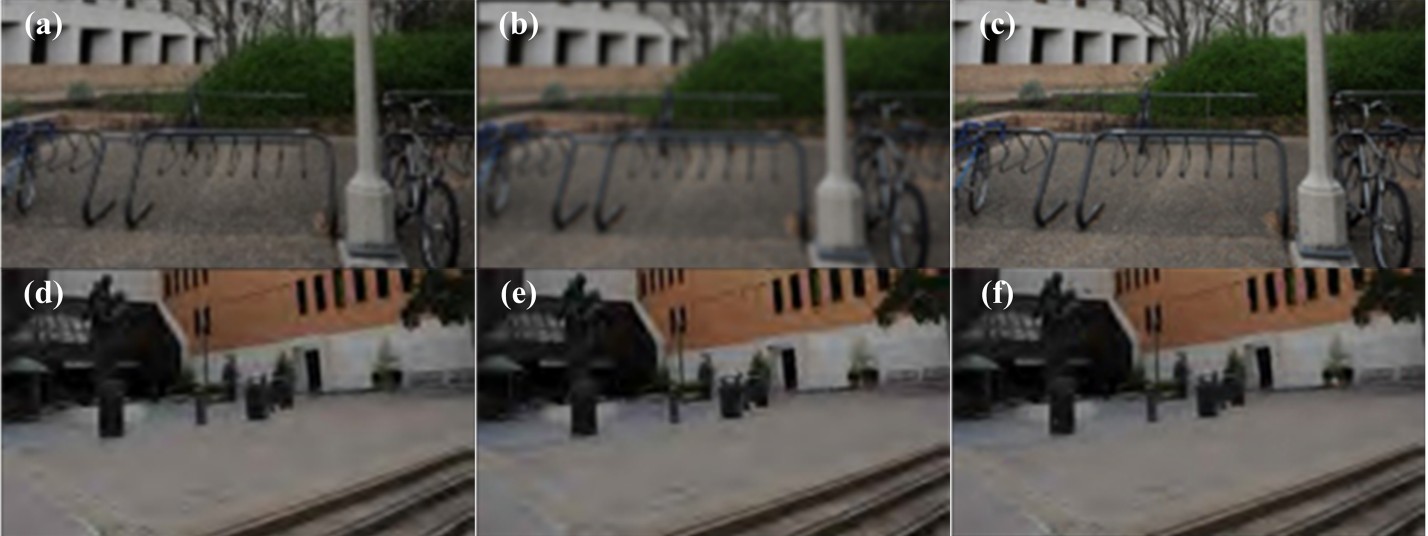

**Figure 2 Left and right view and RGB fused view of the stereo image.** (A) Reference left view; (B) BLUR distortion right view; (C) RGB fusion view; (D) JP2K distorted left view; (E) JP2K distortion right view; (F) JP2K distorted RGB fusion view. Figure source credits: (A–C) are from LIVE 3D Image Quality Database—Phase I. (D–F) are from LIVE 3D Image Quality Database—Phase II.

represent the spatial center frequencies of the complex sinusoidal grating. $\theta$ indicates the orientation of the filter.

The examples of synthesized RGB fusion views under symmetric and asymmetric distortion scenarios are shown in Fig. 2. As shown in Fig. 2A represents the undistorted reference left view, Fig. 2B represents the right view with Gaussian blur (BLUR) distortion, and Fig. 2C represents the RGB fusion view synthesized using the MB-FR-SIQA method. In Fig. 2C, the impact of noise is almost imperceptible. This is because Fig. 2A is undistorted, with higher contrast and clearer contours. Figure 2B is distorted, resulting in lower contrast and less distinct contours. Therefore, during the binocular competition process, Fig. 2A dominates the competition, leading to minimal noise in Fig. 2C. This demonstrates that the RGB fusion view can effectively reflect the binocular competition phenomenon. Panels Figs. 2D and 2E are examples of RGB fusion views synthesized from symmetrically JP2K distorted images using the MB-FR-SIQA method.

## Extract binocular features

The RGB fusion view is generated through the complex process of binocular fusion and binocular competition in the human brain. Extracting effective feature information from the RGB fusion view can reflect the binocular quality. In this section, local features are first extracted, including local phase and local amplitude features. Then, the saliency map of the RGB fusion view is computed, which is used to adjust the weights of the local features. Finally, global features are extracted, including brightness, contrast, gradient magnitude, and phase consistency (PC) features.

## Extract local features

Phase is an important feature in image quality assessment as it can capture a significant portion of the positional and structural information within an image. It exhibits high stability and adaptability and finds wide applications in image processing (*Qian & Mikaelian, 2000*). However, the global phase obtained through Fourier transform cannot effectively express local features. Log-Gabor filters can mimic the simple cell part of the HVS cortex in this article (*Field, 1987*). The log-Gabor filters maximize the fourier components to calculate the PC between the reference RGB fusion view and the distorted RGB fusion view. The formula for computing the energy response $G_{s,o}(\omega, \theta)$ in the Fourier domain with four scales and four directions is given by:

$$G_{s,o}(\omega, \theta) = \exp\left[-\frac{(\log(\omega/\omega_s))^2}{2\sigma_s^2}\right] \times \left[-\frac{(\theta - \theta_o)^2}{2\sigma_o^2}\right] \tag{5}$$

where $\omega$ is the radial coordinate. $\theta$ is the angular coordinate. $s$ is the spatial scale index. $o$ is the spatial orientation index. $\omega_s$ is the center frequency of the filter. $\theta_o$ is the orientation of the filter. Parameters $\sigma_s$ and $\sigma_o$ are used to control the strength of the filter.

A set of energy responses of the log-Gabor filter in the $o$ direction and $s$ scale is expressed as: $[\eta_{s,o}(x), \xi_{s,o}(x)]$. Then the local amplitude $LA_{s,o}(x)$ at position $x$ is calculated as follows:

$$LA_{s,o}(x) = \sqrt{\eta_{s,o}(x)^2 + \xi_{s,o}(x)^2}. \tag{6}$$

The local energy $E_o(x)$ along the $o$ direction at position $x$ is calculated as follows:

$$\begin{aligned}
E_o(x) &= \sqrt{(H_o(x))^2 + (F_o(x))^2} \\
H_o(x) &= \sum_s \eta_{s,o}(x) \\
F_o(x) &= \sum_s \xi_{s,o}(x).
\end{aligned} \tag{7}$$

The phase consistency $PC_o(x)$ along the $o$ direction at position $x$ is calculated as follows:

$$PC_o(x) = \frac{E_o(x)}{\sum_s A_{s,o}(x) + \varepsilon} \tag{8}$$

where $\varepsilon$ is a very small positive value. The global phase can not effectively express the local features. We use the local phase and local amplitude to describe the local features. The calculation formula for the local phase $LP_{local}(x)$ at position $x$ is defined as the angle along the direction of the maximum PC value between $F_o(x)$ and $H_o(x)$, and it is given by:

$$LP_{local}(x) = \arctan(H_{o_m}(x), F_{o_m}(x)) \tag{9}$$

where $o_m$ represents the direction corresponding to the maximum PC value. The local amplitude at point $x$ is denoted as $LA_{local}(x)$. It is defined as the sum of local

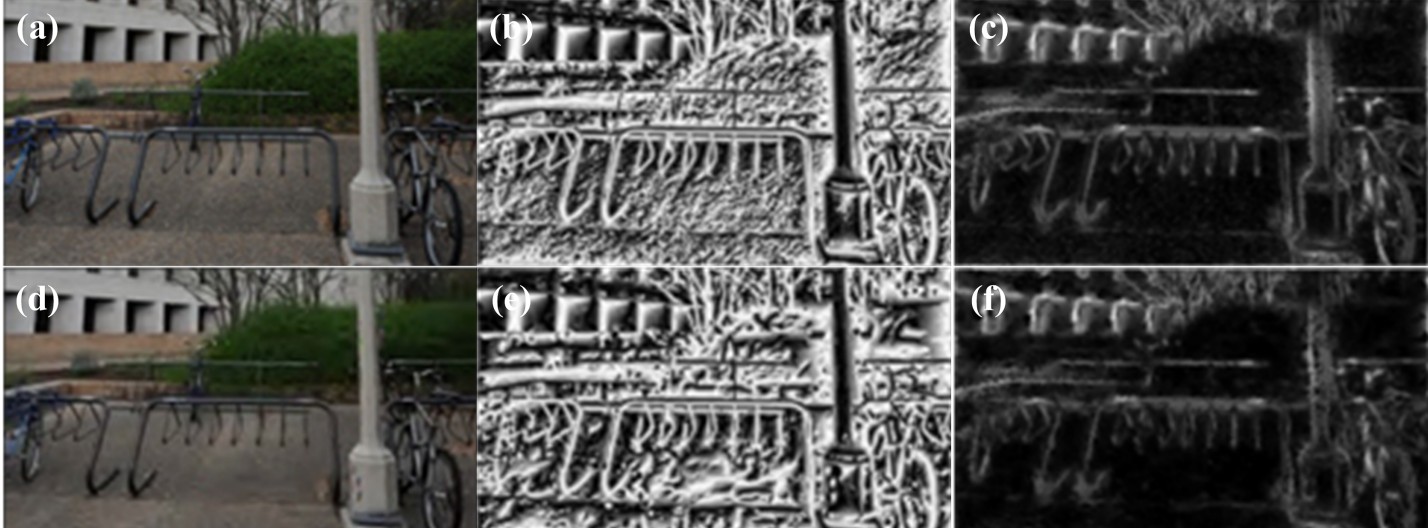

**Figure 3  RGB fusion view, local phase map, local amplitude map.** (A) Reference RGB fusion view; (B) reference local phase map; (C) reference local amplitude map; (D) JP2K distorted RGB fusion view; (E) JP2K distortion local phase map; (F) JP2K distortion local amplitude map. Figure source credits: (A and D) are from LIVE 3D Image Quality Database—Phase I. (B and C) are from LIVE 3D Image Quality Database—Phase II. (E and F) were generated using the code from LIVE 3D Image Quality Database—Phase II.               

amplitudes across all scales along the $o_m$ direction. The calculation formula is shown as follows:

$$LA_{local}(x) = \sum_s LA_{s,o_m}(x). \tag{10}$$

In this article, the log-Gabor filter is configured with four scales and four orientations. The parameter settings are as follows: $\omega_o = 1/6$, $\theta_o = 0$, $\sigma_r = 0.3$, $\sigma_\theta = 0.4$. By using RGB fusion visualization, the resulting local phase and local amplitude images are shown in Fig. 3.

### Saliency map

Different positions in an image contribute differently to the perception of image quality. To represent the visual importance of local regions, we use saliency maps. The saliency map is computed using an improved method called SDSP (*Zhang, Gu & Li, 2013*). Since the human eye is sensitive to changes in image edge positions, edge saliency is used instead of center saliency. First, people are more interested in warm colors. Second, bandpass filters can capture object features in visual scenes. Finally, human eyes are more sensitive to the perception of edge regions than smooth regions. The saliency map obtained by the method in this article clearly shows the shape and boundaries of objects. The saliency map is defined as $VS(x)$, and the saliency map is calculated as follows:

$$VS(x) = S_F(x) \cdot S_E(x) \cdot S_C(x) \tag{11}$$

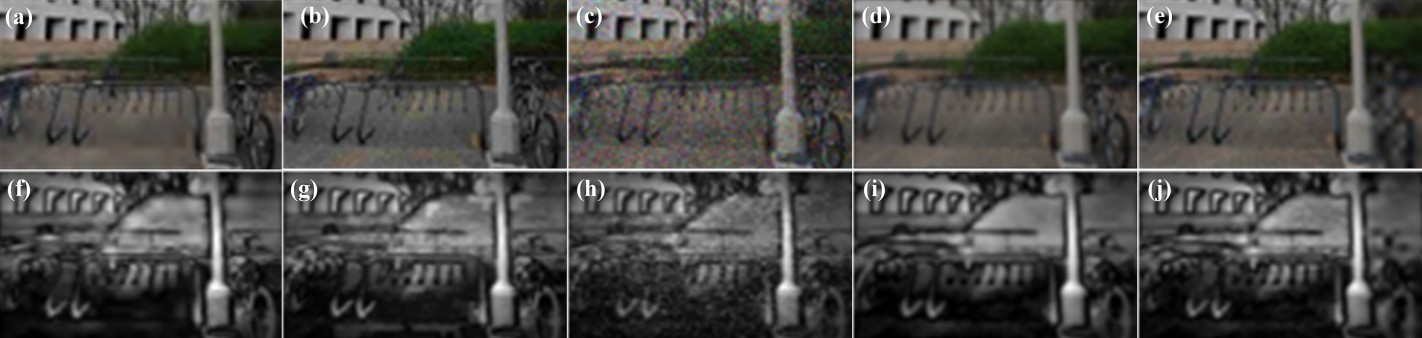

**Figure 4 Saliency map based on RGB fused views with different distortion types.** (A) JP2K distortion map; (B) JPEG distortion map; (C) WN distortion map; (D) BLUR distortion map; (E) FF distortion map; (F) JP2K distortion saliency map; (G) JPEG distortion salience map; (H) WN distortion salience map; (I) BLUR distortion saliency map; (J) FF distortion salience map. Figure source credits: (A–E) are from LIVE 3D Image Quality Database–Phase I. (F–J) are from LIVE 3D Image Quality Database–Phase II.

where $S_F(x)$ is the saliency map at position $x$. $S_E(x)$ is the edge saliency map at position $x$. $S_C(x)$ is the color saliency map at position $x$. Based on the different types of distortions, the saliency maps generated through RGB fusion visualization are shown in Fig. 4.

## Extract global features

The global features extracted based on the RGB fusion view include brightness, contrast, gradient magnitude, and PC. Among them, brightness and contrast represent the distortion of stereo image content, while gradient magnitude and PC represent the distortion of stereo image structure.

According to *Wang et al. (2004)*, we define the brightness feature similarity at position $x$ as $S_L(x)$ and the contrast feature similarity as $S_C(x)$. Their formulas for calculation are as follows:

$$S_L(x) = \frac{2\mu_x\mu_y + C_1}{\mu_x^2 + \mu_y^2 + C_1} \tag{12}$$

$$S_C(x) = \frac{2\sigma_x\sigma_y + C_2}{\sigma_x^2 + \sigma_y^2 + C_2} \tag{13}$$

where, $x$ and $y$ represent the reference image and the distorted image, respectively. $\mu_x$ and $\mu_y$ represent the mean values of $x$ and $y$. $\sigma_x$ and $\sigma_y$ represent the standard deviations of $x$ and $y$. $\sigma_{xy}$ represents the covariance between $x$ and $y$. $C_1$ and $C_2$ are constants used to avoid errors caused by division by zero in the denominators.

The gradient magnitude is an effective way to capture the structural features of an image. We use the Prewitt operator (*Nezhadarya & Ward, 2009*) to obtain the gradient map of the RGB fusion view. The gradient magnitude represents the structural

information of the stereo image. The gradient magnitude $GM(x)$ at position $x$ is calculated as follows:

$$GM(x) = \sqrt{G_x(x)^2 + G_y(x)^2}$$

$$G_x(x) = \frac{1}{3} \begin{bmatrix} 1 & 0 & -1 \\ 1 & 0 & -1 \\ 1 & 0 & -1 \end{bmatrix} \otimes f(x) \tag{14}$$

$$G_y(x) = \frac{1}{3} \begin{bmatrix} 1 & 1 & 1 \\ 0 & 0 & 0 \\ -1 & -1 & -1 \end{bmatrix} \otimes f(x)$$

where $\otimes$ represents the convolution operation. $f(x)$ represents the image. $G_x(x)$ and $G_y(x)$ represent the horizontal and vertical gradients.

The formula for calculating the similarity of gradient magnitudes $S_{GM}(x)$ at position $x$ is shown as follows:

$$S_{GM}(x) = \frac{2GM_{ref}(x) \times GM_{dis}(x)}{GM_{ref}(x)^2 + GM_{dis}(x)^2 + C_3} \tag{15}$$

where $C_3$ is a constant used to prevent division by zero. $GM_{ref}(x)$ and $GM_{dis}(x)$ represent the gradient magnitude of the RGB reference and distorted fusion views.

PC provides relative invariance to image variations and facilitates the extraction of stable features in the image. The calculation formula of the similarity $S_{PC}(x)$ of PC at position $x$ is shown as follows:

$$S_{PC}(x) = \frac{2PC_{ref}(x) \times PC_{dis}(x)}{PC_{ref}(x)^2 + PC_{dis}(x)^2 + C_4} \tag{16}$$

where $C_4$ is a constant used to prevent division by zero. $PC_{ref}(x)$ and $PC_{dis}(x)$ represent the PC of the RGB reference and distorted fusion views.

**Extract monocular features**

In stereo images, the feature information from the left and right views can effectively reflect the quality of the stereo image. According to Formulas (12), (13), and (15), the brightness similarity, contrast similarity, and gradient magnitude similarity of the left and right views are calculated respectively. Then effectively fuse the above three feature similarities. Taking the left view as an example, the formula for calculating the similarity $S_l(x)$ between the reference left view and the distorted left view at position $x$ is shown as follows:

$$S_l(x) = [S_L(x)]^\alpha \cdot [S_C(x)]^\beta \cdot [S_{GM}(x)]^\gamma \tag{17}$$

where $\alpha$, $\beta$, and $\gamma$ are the balance parameters for adjusting brightness, contrast, and gradient magnitude, respectively. We consider all three factors to be equally important. Set $\alpha = \beta = \gamma = 1$. Similarly, we calculate the feature similarity $S_r(x)$ for the right view.

**Peer**J Computer Science

## Quality assessment based on monocular and binocular features

When observing stereo images, both monocular and binocular vision phenomena occur simultaneously. Therefore, fusing monocular and binocular features improves the predictive ability of image quality. We calculate monocular and binocular quality scores based on their respective features. And we propose a merging method to combine them into an overall quality score for stereo images. The quality assessment process is divided into four steps, which are detailed as follows.

The first step is to extract global and local features based on the RGB fusion view, which effectively represents the binocular quality. For the local phase and local amplitude features, similarity can be computed using a method similar to Eq. (15). Then the local phase feature similarity $S_{LP}(x)$ and local amplitude feature similarity $S_{LA}(x)$ are calculated as follows:

$$S_{LP}(x) = \frac{2LP_{local}^{ref}(x) \cdot LP_{local}^{dis}(x) + \varepsilon_1}{\left(LP_{local}^{ref}(x)\right)^2 + \left(LP_{local}^{dis}(x)\right)^2 + \varepsilon_1} \tag{18}$$

$$S_{LA}(x) = \frac{2LA_{local}^{ref}(x) \cdot LA_{local}^{dis}(x) + \varepsilon_2}{\left(LA_{local}^{ref}(x)\right)^2 + \left(LA_{local}^{dis}(x)\right)^2 + \varepsilon_2} \tag{19}$$

where $\varepsilon_1$ and $\varepsilon_2$ are constants used to prevent division by zero. $LP_{local}^{ref}(x)$ and $LA_{local}^{ref}(x)$ represent the local phase and local amplitude features extracted from the reference RGB fusion view. $LP_{local}^{dis}(x)$ and $LA_{local}^{dis}(x)$ represent the local phase and local amplitude features extracted from the distorted RGB fusion view. To comprehensively combine the local features, we integrate the local phase and local amplitude to form an overall local quality score $S_1(x)$, which is given by:

$$S_1(x) = W_{LP} \cdot S_{LP}(x) + W_{LA} \cdot S_{LA}(x) \tag{20}$$

where $W_{LP}$ is the weight for the local phase. $W_{LA}$ is the weight for the local amplitude. The experimental results show that in the LIVE 3D Phase I database (LIVE P-I), $W_{LP} = 0.6$ and $W_{LA} = 0.4$ yield the best results. In the LIVE 3D Phase II (LIVE P-II) database, $W_{LP} = 0.75$ and $W_{LA} = 0.25$ yield the best results.

For a given position $x$, if either the reference or distorted RGB fusion view has a high saliency value, then position $x$ has a high impact on the HVS when evaluating the similarity between the two images (*Henriksson, Hyvärinen & Vanni, 2009*). Therefore, we weight $S_1(x)$ with higher saliency values in the reference and warped RGB-fused views to obtain an optimized binocular local feature quality score $Q_1$, calculated as follows:

$$Q_1 = \frac{\sum_{x \in \Omega} S_1(x) \cdot VS_m(x) \cdot S_{VS}(x)}{\sum_{x \in \Omega} VS_m(x)}$$

$$S_{VS}(x) = \frac{2VS_{ref}(x) \cdot VS_{dis}(x) + \varepsilon_3}{VS_{ref}^2(x) + VS_{dis}^2(x) + \varepsilon_3} \tag{21}$$

$$VS_m(x) = \max(VS_1(x), VS_2(x))$$

where $\Omega$ is the spatial domain of the image. $VS_m(x)$ represents the higher saliency value between the reference and distorted RGB fusion views. $S_{VS}(x)$ represents the feature similarity of the saliency map. $VS_{ref}(x)$ and $VS_{dis}(x)$ are the saliency maps of the reference and distorted RGB-fused views. $\varepsilon_3$ is a constant to prevent division by zero.

In the second step, global features are extracted based on the RGB fusion view, including brightness, contrast, gradient magnitude, and PC features. The gradient magnitude similarity is calculated using Eq. (15), and the PC similarity is calculated using Eq. (16). Then, the gradient magnitude and PC are effectively fused, representing the structural similarity $S_2(x)$. Calculated as follows:

$$S_2(x) = S_{PC}(x) \cdot S_{GM}(x) \tag{22}$$

To obtain accurate structural similarity, we use the larger PC values from the reference and distorted RGB fusion views to weigh the overall structural similarity:

$$S_3(x) = \frac{\sum_{x \in \Omega} S_2(x) \cdot PC_m(x)}{\sum_{x \in \Omega} PC_m(x)} \tag{23}$$
$$PC_m(x) = \max(PC_{ref}(x), PC_{dis}(x))$$

where $\Omega$ is the spatial domain of the image. $PC_m(x)$ indicates higher PC values in reference and distorted RGB fused views.

Calculate the structural similarity $S_3(x)$ according to Eq. (23). Calculate brightness similarity $S_L(x)$ and contrast similarity $S_C(x)$ according to Eqs. (12) and (13). The MS-SSIM (*Wang, Simoncelli & Bovik, 2003*) method is used to calculate the binocular quality score $Q_2$ for the three types of similarity information:

$$Q_2 = \sum_{x \in C} \frac{Q_{MS-SSIM}(C_{ref}(x), C_{dis}(x))}{N_C} \tag{24}$$

where $N_C$ represents the number of pixels in the RGB fused view. $C_{ref}(x)$ and $C_{dis}(x)$ represent the original fused view and the distorted fused view.

In the third step, we calculate the quality score $S_l(x)$ of the left view according to Eq. (17). And we calculate the quality score $S_r(x)$ of the right view in the same way. Then the calculation of monocular quality score $Q_3$ is given by:

$$Q_3 = \frac{S_l(x) + S_r(x)}{2} \tag{25}$$

In the fourth step, the final quality prediction score $Q$ is calculated by fusing the monocular and binocular quality scores. The overall quality score $Q$ is calculated as follows:

$$Q = aQ_1 + bQ_2 + cQ_3 \tag{26}$$

where $a$, $b$, and $c$ are different weights assigned to $Q_1$, $Q_2$, and $Q_3$, and $a + b + c = 1$. During the experimental process, different parameter combinations are used to achieve the best

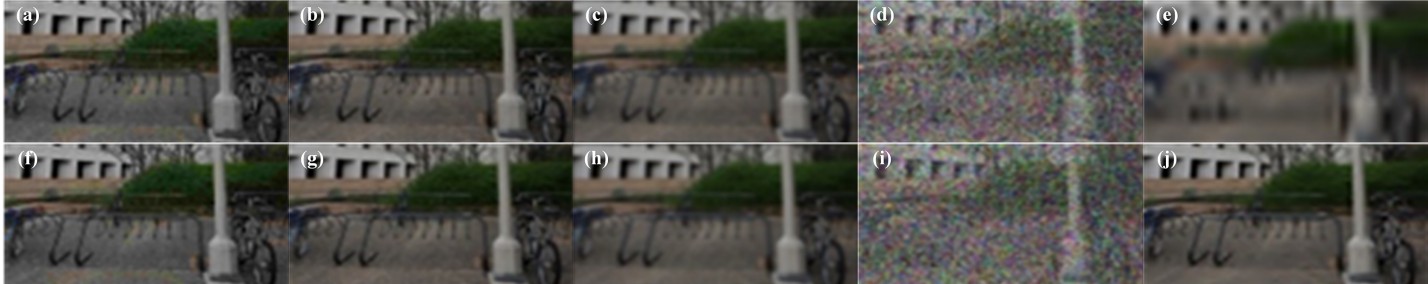

**Figure 5 Stereo image pairs with different distortion types from the LIVE P-I database.** (A) JPEG distorted left view; (B) JP2K distorted left view; (C) BLUR distorted left view; (D) WN distorted left view (E); FF distorted left view; (F) JPEG distorted right view, (G) JP2K distorted right view, (H) BLUR distorted right view, (I) WN distorted right view, (J) FF distorted right view. Figure source credits: (A–J) are from LIVE 3D Image Quality Database—Phase I.

results. In the LIVE P-I, the best-predicted quality score is obtained when $a = 0.5$, $b = 0.2$, and $c = 0.3$. In the LIVE P-II, the best-predicted quality score is obtained when $a = 0.6$, $b = 0.2$, and $c = 0.2$.

## Results and analysis

This section provides a comprehensive assessment of the performance of the method. Firstly, it describes the databases and performance metrics used in the experiments. Secondly, it presents a comparative analysis of the overall performance of the method against other relevant methods, as well as performance comparisons for each type of distortion. Then perform consistency performance prediction and ablation experiments and time complexity analysis.

### SIQA databases

We evaluate the effectiveness and robustness of the MB-FR-SIQA method on two publicly vailable databases (*Ye et al., 2012*).

LIVE P-I database consists of 20 reference stereo image pairs and 365 symmetrically distorted stereo image pairs. All stereo image pairs exhibit symmetric distortions, meaning that the left and right views have the same types and degrees of distortion. The distortion types include JPEG, JPEG2000 (JP2K), White Noise (WN), Gaussian Blur (BLUR), and Fast Fading (FF). There are 45 image pairs with WN distortion, and each of the other distortion types has 80 image pairs.

LIVE P-II database contains both symmetric and asymmetrically distorted stereo image pairs. It comprises a total of eight reference image pairs and 360 distorted image pairs. The distortion types are the same as those in the LIVE P-I database. Each distortion type contains 72 image pairs. The subjective quality scores in both databases are represented by Differential DMOS. The smaller the DMOS value, the smaller the difference and the better the quality. Figure 5 shows the left and right views of different distortion types in the LIVE P-I database.

## Performance assessment index

To effectively evaluate the performance of the MB-FR-SIQA method. We adopt three commonly used assessment metrics to evaluate its performance: Pearson Linear Correlation Coefficient (PLCC), Spearman Rank Order Correlation Coefficient (SROCC), and Root Mean Square Error (RMSE). PLCC assesses the accuracy of the method. SROCC judges the monotonicity of the method and RMSE reflects the consistency of the method. The closer PLCC and SROCC are to 1, and the closer RMSE is to 0, the better the prediction performance of the method. For nonlinear regression between subjective and objective scores, we use the five-parameter logistic function (*Sheikh, Sabir & Bovik, 2006*) to compute:

$$f(x) = \beta_1 \times \left[ \frac{1}{2} - \frac{1}{1 + \exp(\beta_2(x - \beta_3))} \right] + \beta_4 x + \beta_5 \tag{27}$$

Among them, $x$ represents the predicted score. $f(x)$ is the mapped score. $\beta_1, \beta_2, \beta_3, \beta_4, \beta_5$ are parameters determined by the best fit between subjective and objective scores.

## Method performance

The experiments are conducted in two stages on the LIVE 3D IQA database to validate the effectiveness and consistency of the proposed BM-FR-SIQA method in this article. Firstly, the performance of local feature extraction, overall performance evaluation, and performance evaluation for a single type of distortion is being verified. Then, consistency performance prediction and ablation experiments are being conducted, followed by a summary of this chapter.

## Local feature extraction performance

Feature extraction is one of the most critical procedures in SIQA. Better feature extraction results not only help to improve the accuracy of the SIQA method but also help to coincide with subjective results. Our experiment uses four different schemes in the LIVE P-I and LIVE P-II databases to demonstrate the effectiveness of feature extraction. All schemes are based on RGB fused views, and their basic composition is as follows: Scheme 1 only uses local phases to calculate local feature scores. Scheme 2 uses only local amplitudes to compute local mass fractions. Scheme 3 uses a combination of local phase and amplitude to calculate the mass fraction. Scheme 4 is the MB-FR-SIQA method, which uses the saliency map to weigh the local phase and amplitude to calculate the quality score. The overall performance comparison results of the four schemes and the MB-FR-SIQA method are presented in Table 1. Compared with Schemes 1 and 2, Scheme 3 uses a single local phase or local amplitude which is not as effective as fusing local phase and amplitude. Compared with Scheme 3, Scheme 4 is effective in using saliency-weighted local features.

## Overall performance assessment

In the LIVE P-I database, all images are symmetric, whereas 66% of the images in the LIVE P-II database exhibit asymmetric distortions. To investigate the performance of the proposed method on symmetric and asymmetric distortion databases, experiments are

**Table 1 Overall performance comparison under different schemes.**

| | LIVE P-I | | | | LIVE P-II | | | |
|---|---|---|---|---|---|---|---|---|
| Scheme | Scheme 1 | Scheme 2 | Scheme 3 | Ours | Scheme 1 | Scheme 2 | Scheme 3 | Ours |
| PLCC | 0.9204 | 0.9232 | 0.9357 | **0.9388** | 0.9197 | 0.8901 | 0.9241 | **0.9245** |
| SROCC | 0.9143 | 0.9172 | 0.9284 | **0.9307** | 0.9099 | 0.8760 | 0.9160 | **0.9174** |
| RMSE | 6.4122 | 6.3032 | 5.7855 | **5.6494** | 4.4318 | 5.1447 | 4.3118 | **4.3017** |

**Note:**
The best-performing is displayed in the bold.

**Table 2 Overall performance comparison in LIVE 3D IQA database.**

| Type | Method | LIVE P-I | | | LIVE P-II | | |
|---|---|---|---|---|---|---|---|
| | | PLCC | SROCC | RMSE | PLCC | SROCC | RMSE |
| Traditional method | *Wang et al. (2004)* | 0.8766 | 0.8765 | 7.8913 | 0.8024 | 0.7925 | 6.7366 |
| | *Wang, Simoncelli & Bovik (2003)* | 0.9261 | 0.9239 | 8.2486 | 0.7824 | 0.7774 | 8.0217 |
| | *Chen et al. (2013)* | 0.8910 | 0.8950 | 7.2470 | 0.8800 | 0.8950 | 5.1020 |
| NR | *Oh et al. (2017)* | 0.9430 | 0.9360 | – | 0.8630 | 0.8710 | – |
| | *Chen, Cormack & Bovik (2013)* | 0.939 | 0.930 | 5.605 | **0.922** | **0.913** | **4.352** |
| | *Benoit, Le Callet & Campisi (2008)* | 0.8899 | 0.8901 | 7.4786 | 0.7642 | 0.7475 | 7.2806 |
| | *Lin & Wu (2014)* | 0.8645 | 0.8559 | 8.2424 | 0.6584 | 0.6375 | 8.4956 |
| | *You et al. (2010)* | 0.9303 | 0.9247 | 6.0161 | 0.7744 | 0.7206 | 7.1413 |
| FR | *Chen et al. (2013)* | 0.9167 | 0.9157 | 6.5503 | 0.9010 | 0.8930 | 4.9870 |
| | *Li et al. (2015)* | 0.9228 | 0.9136 | – | 0.8257 | 0.8059 | – |
| | *Chen et al. (2013)* | 0.9245 | 0.9217 | 6.2522 | 0.7585 | 0.7451 | 7.3554 |
| | *Liu, Kong & Zhen (2019)* | 0.9430 | **0.9402** | 5.5423 | 0.8417 | 0.8317 | 6.0946 |
| | Ours | **0.9454** | 0.9381 | **5.3465** | 0.9101 | 0.9021 | 4.6778 |

**Note:**
The symbol '–' indicates unavailable results, and the best-performing method is highlighted in bold.

conducted in two phases of the LIVE 3D IQA database in this article. To comprehensively evaluate the performance of the MB-FR-SIQA method, we compare the performance of the MB-FR-SIQA method with 10 existing SIQA methods including three NR-SIQA methods and seven FR-SIQA methods.

Table 2 shows the overall performance of the different approaches in the above two databases. In Table 2, it can be observed that in both databases, traditional methods like SSIM (*Wang et al., 2004*) and MS-SSIM (*Wang, Simoncelli & Bovik, 2003*) perform poorly compared to the MB-FR-SIQA method due to their insufficient consideration of stereo characteristics. *Benoit, Le Callet & Campisi (2008)* heavily depended on stereo matching. *Chen et al. (2013)* constructed fused views from reference and distorted stereo image pairs and then used 2D-IQA methods to evaluate stereo image quality. However, due to the differences between 2D and 3D images, the assessment results may not be accurate. *Liu, Kong & Zhen (2019)* improved the predictive performance of SIQA to some extent by considering the binocular interaction process of the HVS, but they ignored the influence of features from the left and right views on quality assessment. *Shao et al. (2013)* considered

**Table 3 Comparison results of PLCC values of different distortion types in the LIVE 3D IQA database.**

| Type | Method | LIVE P-I | | | | | LIVE P-II | | | | |
|---|---|---|---|---|---|---|---|---|---|---|---|
| | | JP2K | JPEG | WN | BLUR | FF | JP2K | JPEG | WN | BLUR | FF |
| NR | *Chen et al. (2013)* | 0.9070 | 0.6950 | 0.9170 | 0.9510 | 0.7350 | 0.8990 | **0.9010** | 0.9470 | 0.9410 | 0.9320 |
| | *Oh et al. (2017)* | 0.9130 | **0.7670** | 0.9100 | 0.9500 | 0.9540 | 0.8650 | 0.8210 | 0.8360 | 0.9430 | 0.8150 |
| | *Chen, Cormack & Bovik (2013)* | 0.926 | 0.668 | 0.941 | 0.935 | 0.845 | 0.835 | 0.859 | 0.953 | 0.978 | **0.925** |
| FR | *Benoit, Le Callet & Campisi (2008)* | 0.8897 | 0.5597 | 0.9360 | 0.9256 | 0.7514 | 0.6470 | 0.5530 | 0.8610 | 0.8810 | 0.8470 |
| | *Lin & Wu (2014)* | 0.8381 | 0.6654 | 0.9280 | 0.8249 | 0.7086 | 0.8430 | 0.8620 | 0.9570 | 0.9630 | 0.9010 |
| | *You et al. (2010)* | 0.9410 | 0.6333 | 0.9351 | 0.9545 | 0.8589 | 0.7320 | 0.6741 | 0.5464 | 0.9763 | 0.8561 |
| | *Chen et al. (2013)* | 0.9166 | 0.6356 | 0.9353 | 0.9418 | 0.7865 | 0.8426 | 0.8422 | **0.9602** | 0.9605 | 0.9097 |
| | *Li et al. (2015)* | 0.9272 | 0.6323 | 0.9221 | **0.9564** | **0.8913** | 0.8206 | 0.7527 | 0.8464 | 0.609 | 0.9030 |
| | *Chen et al. (2013)* | 0.9238 | 0.6563 | 0.9410 | 0.9513 | 0.8403 | 0.8377 | 0.7504 | 0.8496 | 0.8270 | 0.8808 |
| | *Liu, Kong & Zhen (2019)* | 0.9423 | 0.7315 | 0.9463 | 0.9530 | 0.8658 | 0.8701 | 0.8758 | 0.9325 | 0.9430 | 0.9218 |
| | Ours | **0.9478** | 0.7407 | **0.9576** | 0.9513 | 0.8369 | **0.8771** | 0.8823 | 0.9515 | **0.9777** | 0.8587 |

**Note:**
The best-performing is displayed in the bold.

both global and local features, but some other important HVS features were neglected. Among NR methods, *Messai, Hachouf & Seghir (2020)* also obtained fused views and disparity maps from stereo image pairs, but only extracted gradient features related to them. They neglected other important visual features. Due to the superiority of CNN networks in feature extraction, *Oh et al. (2017)* performed better than other NR methods in the Phase I database, but it failed to achieve the expected performance in the Phase II database due to the lack of stereo features.

Overall, the method proposed in this article achieves optimal performance in terms of PLCC and RMSE on the LIVE P-I database, with SROCC also ranking among the top. On the LIVE P-II database, the addition of asymmetric distortions to stereo images introduces binocular rivalry during observation. However, the RGB fusion view employed in this article effectively mitigates the binocular rivalry caused by asymmetric distortions. Consequently, the performance metrics of the MB-FR-SIQA method on the LIVE P-II database surpass those of other methods.

### Single distortion type performance assessment

An effective SIQA method should not only demonstrate overall accuracy across the entire database but also be capable of predicting quality for individual distortion types. In this section, we conduct separate experiments for each distortion type in the LIVE P-I and LIVE P-II databases to validate the generality of the proposed method. The PLCC and SROCC values for each distortion type are in different databases in Tables 3 and 4, including JP2K, JPEG, WN, FF, and BLUR. The MB-FR-SIQA method exhibits the best performance in predicting JP2K and WN distortion types in the Phase I database, while in the Phase II database, it performes best in predicting JP2K and BLUR distortion types. For other distortion types, the MB-FR-SIQA method maintains competitive performance, showing certain advantages over the comparative methods.

**Table 4 Comparison results of SROCC values of different distortion types in the LIVE 3D IQA database.**

| Type | Method | LIVE P-I | | | | | LIVE P-II | | | | |
|------|--------|----------|---|---|---|---|-----------|---|---|---|---|
| | | JP2K | JPEG | WN | BLUR | FF | JP2K | JPEG | WN | BLUR | FF |
| NR | *Chen et al. (2013)* | 0.8630 | 0.6170 | 0.9190 | 0.8780 | 0.6520 | 0.8670 | **0.8670** | 0.9500 | 0.9000 | **0.9330** |
| | *Oh et al. (2017)* | 0.8850 | 0.7620 | 0.9210 | 0.9300 | **0.9440** | 0.8530 | 0.8220 | 0.8330 | 0.8890 | 0.8780 |
| | *Chen, Cormack & Bovik (2013)* | 0.899 | 0.625 | 0.941 | 0.887 | 0.777 | 0.842 | 0.837 | 0.943 | 0.913 | 0.925 |
| FR | *Benoit, Le Callet & Campisi (2008)* | 0.8701 | 0.5186 | 0.9147 | 0.8967 | 0.6142 | 0.6320 | 0.5070 | 0.8650 | 0.8540 | 0.8310 |
| | *Lin & Wu (2014)* | 0.8388 | 0.1960 | 0.9284 | 0.7910 | 0.6581 | 0.8140 | 0.8430 | 0.9502 | 0.9096 | 0.8840 |
| | *You et al. (2010)* | 0.8979 | 0.5967 | 0.9389 | 0.9278 | 0.8030 | 0.7309 | 0.5229 | 0.4820 | 0.9227 | 0.8392 |
| | *Chen et al. (2013)* | 0.8954 | 0.5632 | 0.9376 | 0.9283 | 0.7391 | 0.8334 | 0.8396 | **0.9554** | 0.9096 | 0.8890 |
| | *Li et al. (2015)* | 0.8925 | 0.6218 | 0.9161 | 0.9368 | 0.8450 | 0.8179 | 0.7072 | 0.8316 | 0.9356 | 0.8059 |
| | *Chen et al. (2013)* | 0.8752 | 0.6148 | 0.9431 | **0.9375** | 0.7814 | 0.8477 | 0.7195 | 0.8455 | 0.8005 | 0.8509 |
| | *Liu, Kong & Zhen (2019)* | 0.9040 | **0.6952** | 0.9468 | 0.9294 | 0.8108 | 0.8642 | 0.8640 | 0.9230 | 0.9114 | 0.8977 |
| | Ours | **0.9079** | 0.6733 | **0.9494** | 0.9264 | 0.7859 | **0.8665** | 0.8417 | 0.9474 | **0.9267** | 0.8468 |

**Note:**
The best-performing is displayed in the bold.

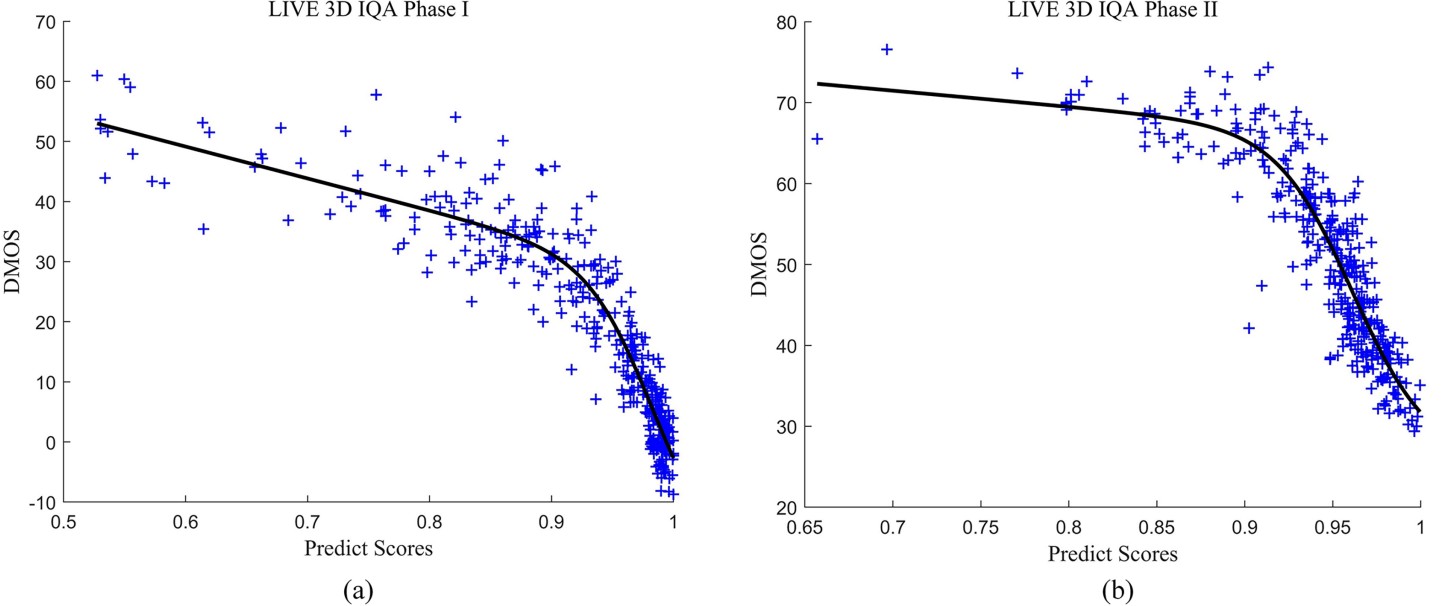

**Figure 6 Scatter diagram of global performance evaluation.** (A) LIVE P-I scatter diagram; (B) LIVE P-II scatter diagram.

## Consistency performance prediction

To provide a more comprehensive analysis of the experimental results, we visualize the overall differences between subjective assessment scores and predicted scores. The scatter plots of the predicted values and DMOS scores for the MB-FR-SIQA method on the LIVE P-I and LIVE P-II databases in Fig. 6. The x-axis represents DMOS scores, while the y-axis represents the predicted quality scores of the distorted images. The closer the scatter points

**Table 5 Ablation experiment results in the LIVE 3D IQA database.**

| Plan | LIVE P-I | | | LIVE P-II | | |
|---|---|---|---|---|---|---|
| | Plan A | Plan B | Ours | Plan A | Plan B | Ours |
| PLCC | 0.9451 | 0.9397 | **0.9454** | 0.7697 | 0.89446 | **0.9101** |
| SROCC | 0.9393 | 0.9314 | 0.9381 | 0.7314 | 0.88501 | **0.9021** |
| RMSE | 5.3544 | 5.6048 | **5.3465** | 720,501 | 4.9886 | **4.6778** |

Note:
The best-performing is displayed in the bold.

**Table 6 Average computation time for image pairs in the LIVE P-I image library.**

| Method | Messai, Hachouf & Seghir (2020) | Lin & Wu (2014) | Liu, Kong & Zhen (2019) | Ours |
|---|---|---|---|---|
| Average calculation time(s) | 72.5238 | 40.7955 | 13.3999 | 22.0610 |

are to the fitting curve, the higher the consistency between the predicted values and subjective assessment scores. Based on the distribution of scatter points in the figure, it can be observed that the subjective and objective scores tend to be close, indicating that the MB-FR-SIQA method demonstrates good subjective consistency.

## Ablation experiment

We use ablation experiments to verify the effectiveness of the MB-FR-SIQA method. Plan A involves extracting monocular features only for quality score calculation, Plan B trains by solely utilizing binocular features for quality score calculation, Plan C calculates quality scores by fusing both monocular and binocular features. Table 5 presents the comparison of results under different plans. The best-performing is displayed in the bold. Experimental results indicate that combining monocular and binocular visual features can provide a more reasonable and accurate quality assessment method.

## Time complexity analysis

Generally, time complexity is used to describe the running time and efficiency of an algorithm. In order to prove the advantages of this method in terms of time complexity, a comparative experiment is conducted in this section. The experimental environment is: Windows10 system, 11th Gen Inter(R) Core(TM) i5-11400H @ 2.70GHz processor, 16GB RAM, Matlab 2018b. Table 6 shows the average quality score calculation time of each stereo image pair for different methods on LIVE P-I. As can be seen from the table, compared with the methods proposed by *Messai, Hachouf & Seghir (2020)* and *Lin & Wu (2014)*, our method has lower time complexity, which shows that our method has excellent performance in terms of running speed. For the method proposed by *Messai, Hachouf & Seghir (2020)*, it takes the longest time because it uses machine learning methods for training and testing. For the method proposed by *Lin & Wu (2014)*, the feature extraction process is very complex, so the calculation time is doubled compared to the method in this article. Although the method proposed by *Liu, Kong & Zhen (2019)* has a shorter runtime,

its performance is lower than that of the method presented in this article due to the neglect of monocular features.

## CONCLUSIONS

Compared with flat images, stereoscopic images involve many visual characteristics of the HVS. Based on these characteristics, a full-reference stereoscopic image quality assessment method that combines monocular and binocular features is first proposed. Firstly, a three-channel RGB fusion view is generated by combining the Gabor filter bank response and the disparity map. Secondly, monocular and binocular features are extracted respectively based on the monocular view and RGB fusion view. Then the saliency map is performed on the RGB fused view, and the saliency map is used to adjust the weight of local features in the binocular features. Finally, the monocular and binocular quality scores are calculated based on the monocular and binocular features, and the quality scores of the stereo image prediction are obtained by fusion. Performance testing was performed on the public LIVE P-I and Phase-II databases respectively. Experimental results show that the quality scores predicted by the proposed method have good consistency with the subjective assessment score values.

The MB-FR-SIQA method has certain innovation and practical application value. The next steps will delve deeper into research in two aspects: Firstly, due to the complexity of the HVS, its study is still not sufficient. The next step will involve a deeper exploration of the stereo image formation process of the HVS to investigate how to design methods that better conform to the HVS. Secondly, stereo videos are now widely used in social life, but the stereo image quality evaluation method proposed in this article is not suitable for stereo videos. Considering that stereo videos are combinations of stereo images, the next step will explore the similarities between them, seek to improve methods, and extend them to stereo video evaluation methods.

### Funding
This work was supported by the Science and Technology Research Project of Education Department of Jilin Province (No. JJKH20230851KJ, No. JJKH20200799KJ). The funders had no role in study design, data collection and analysis, decision to publish, or preparation of the manuscript.

### Grant Disclosures
The following grant information was disclosed by the authors:
Science and Technology Research Project of Education Department of Jilin Province: JJKH20230851KJ and JJKH20200799KJ.

### Competing Interests
The authors declare that they have no competing interests.

## Author Contributions

- Xiaojuan Hu conceived and designed the experiments, performed the experiments, analyzed the data, prepared figures and/or tables, authored or reviewed drafts of the article, and approved the final draft.
- Jinxin Bai conceived and designed the experiments, performed the experiments, analyzed the data, performed the computation work, prepared figures and/or tables, authored or reviewed drafts of the article, and approved the final draft.
- Chunyi Chen conceived and designed the experiments, authored or reviewed drafts of the article, and approved the final draft.
- Haiyang Yu conceived and designed the experiments, authored or reviewed drafts of the article, and approved the final draft.

## Data Availability

The code is available in the Supplemental File.

The LIVE 3D Image Quality Database—Phase I is available at https://live.ece.utexas.edu/research/quality/live_3dimage_phase1.html.

The LIVE 3D Image Quality Database—Phase II is available at https://live.ece.utexas.edu/research/quality/live_3dimage_phase2.html.

## Supplemental Information

Supplemental information for this article can be found online at http://dx.doi.org/10.7717/peerj-cs.2083#supplemental-information.

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
