# Peer review of "A full reference quality assessment method with fused monocular and binocular features for stereo images"

_PeerJ Computer Science, doi:10.7717/peerj-cs.2083_

## Round 0.1 · original submission · Major Revisions

Dear authors,
You are advised to critically respond to all comments point by point when preparing a new version of the manuscript and while preparing for the rebuttal letter. Please address all the comments/suggestions provided by the reviewers.

Reviewer 1 has requested that you cite specific references. You may add them if you believe they are especially relevant. However, I do not expect you to include these citations, and if you do not include them, this will not influence my decision.

Kind regards,

**Language Note:** The review process has identified that the English language must be improved. PeerJ can provide language editing services - please contact us at [email protected] for pricing (be sure to provide your manuscript number and title). Alternatively, you should make your own arrangements to improve the language quality and provide details in your response letter. – PeerJ Staff
PCoelho

Reviewer 1 ·

Basic reporting

This paper proposes a full reference stereoscopic image quality assessment method by using monocular and binocular features. Experimental results on LIVE 3D IQA database demonstrate the effectiveness of the proposed metric. However, I would suggest a major revision before this work can be considered for publication. I hope the follow comments will help the authors to improve this manuscript.

Experimental design

Some figures (e.g., Figure 1, Figure 2 and Figure 6) in this manuscript are unclear, it is suggested to improve the quality of these figures.

In this paper, the description of the manuscript is suggested to be expressed in the general tense instead of the past tense. Besides, the word “suggested” in line 17 should be replaces by “propose or develop”. Please read more high-quality papers (e.g., IEEE TIP, TMM)to enhance the description.

Validity of the findings

The limitations of the proposed method should be clearly discussed.

It is necessary to validate the computational complexity of the proposed metric, please add a comparative experiment to evaluate the run-time of feature extraction in terms of different IQA methods.

Additional comments

1. The presentation of manuscript is very important for readers and other researchers. I encourage the authors to have their manuscript proof-edited by a fluent English speaker to improve the level of paper presentation.

2. Some specific academic vocabulary in the manuscript are mistranslated, for example the description “stereo picture” in lines 23, 68, “single and binocular features” in lines 529, 534. They should be replaced by “stereo image” , “monocular and binocular features”, please check the full manuscript.

3. In lines 82-84 of this manuscript, we feel puzzle by this description. Is this the third contribution of this manuscript? Please confirm.

4. In this manuscript, only some stereo IQA methods have been reviewed, it is suggested to review some recently-developed IQA papers in the section of introduction, such as: doi: 10.1109/TMM.2023.3330096, doi: 10.1109/TIM.2023.3306527, doi: 10.1109/TMM.2023.3338412.

Reviewer 2 ·

Basic reporting

1. The English expression is not sufficiently standardized and needs refinement to further enhance the accuracy and standardization of the manuscript's presentation.
2. The references cited in the Introduction section are too old. It is recommended to use literature from the past three years for the review analysis.
3. The clarity of the Figures in the manuscript is low and needs revision.

Experimental design

1. It is suggested to further elaborate on the motivation of this manuscript's research in the Introduction section, in order to introduce the novelty of this manuscript.

Validity of the findings

1. The author introduced many deep learning-based no-reference methods in the related work and mentioned, 'While some deep learning-based NR-SIQA methods have achieved promising results in the literature, they often come with high training time complexity, resulting in lengthy processing times for generating results.' Therefore, it is suggested that the author add time complexity comparison experiments in the experimental section to further demonstrate the advancement of the proposed method.

Reviewer 3 ·

Basic reporting

The authors suggested a full reference stereo quality assessment metric based on earlier no-reference approaches. The measure is based on the cyclopean hypothesis, which takes into account binocular rivalry issues. In my opinion. The manuscript has the potential to be published following some revisions. I appreciate that the authors provided the implementation code.

Experimental design

- A cross-validation experiment is needed for better comparison.
- The authors address the issue of binocular rivalry, although symetric/asymetric evaluation is missing from the experimental results.
- Please discuss why the proposed metric performed well on LIVE II but poorly on LIVE I.

Validity of the findings

- Ablation tests are needed to evaluate the effectiveness of monocular and binocular feature extraction.

Additional comments

- The introduction and related work are mixed up. I respectfully advise combining the two sections into one.
- The introduction/related work requires significant improvement to include more recent studies and explore the issue of binocular rivalry.
- Figures are difficult to follow because to extreme compression and incorrect layout.

---

## Round 0.2 · accepted · Accept

Dear authors, we are pleased to verify that you meet the reviewer's valuable feedback to improve your research.

Thank you for considering PeerJ Computer Science and submitting your work.

Reviewer 1 ·

Basic reporting

This is a revised manuscript, The quality is improved after revision, It can be accepted.

Experimental design

The method is good, and the experiments are sufficient.

Validity of the findings

A novel FR image quality assessment method is proposed.

Additional comments

none.

Reviewer 2 ·

Basic reporting

well written

Experimental design

well written

Validity of the findings

well written

Additional comments

The author has addressed all of my concerns. At this stage, I recommend accepting this manuscript for publication.

Reviewer 3 ·

Basic reporting

The authors addressed to my questions; I propose that all of the answers be thoroughly considered during the article discussion session. However, the paper still need figure and table adjustments.

Experimental design

The authors addressed to my questions; I propose that all of the answers be thoroughly considered during the article discussion session. However, the paper still need figure and table adjustments.

Validity of the findings

The authors addressed to my questions; I propose that all of the answers be thoroughly considered during the article discussion session. However, the paper still need figure and table adjustments.

Additional comments

The authors addressed to my questions; I propose that all of the answers be thoroughly considered during the article discussion session. However, the paper still need figure and table adjustments.